# Temporo-cerebellar connectivity underlies timing constraints in audition

Anika Stockert[1,2], Michael Schwartze[2,3], David Poeppel[4,5], Alfred Anwander[2], Sonja A Kotz[2,3]*

[1]Language and Aphasia Laboratory, Department of Neurology, Leipzig University Hospital, Leipzig, Germany; [2]Department of Neuropsychology, Max Planck Institute for Human Cognitive and Brain Sciences, Leipzig, Germany; [3]Department of Neuropsychology and Psychopharmacology, Faculty of Psychology and Neuroscience, Maastricht University, Maastricht, Netherlands; [4]Department of Neuroscience, Max Planck Institute for Empirical Aesthetics, Frankfurt, Germany; [5]Department of Psychology, New York University, New York, United States

**Abstract** The flexible and efficient adaptation to dynamic, rapid changes in the auditory environment likely involves generating and updating of internal models. Such models arguably exploit connections between the neocortex and the cerebellum, supporting proactive adaptation. Here, we tested whether temporo-cerebellar disconnection is associated with the processing of sound at short timescales. First, we identify lesion-specific deficits for the encoding of short timescale spectro-temporal non-speech and speech properties in patients with left posterior temporal cortex stroke. Second, using lesion-guided probabilistic tractography in healthy participants, we revealed bidirectional temporo-cerebellar connectivity with cerebellar dentate nuclei and crura I/II. These findings support the view that the encoding and modeling of rapidly modulated auditory spectro-temporal properties can rely on a temporo-cerebellar interface. We discuss these findings in view of the conjecture that proactive adaptation to a dynamic environment via internal models is a generalizable principle.

*For correspondence: sonja.kotz@maastrichtuniversity.nl

Competing interest: The authors declare that no competing interests exist.

## Introduction

Current theories of motor control postulate that the cerebellum plays a foundational role in monitoring motor performance and its sensory consequences (*Wolpert et al., 1995*; *Wolpert and Miall, 1996*). This important concept has been extended to anticipatory sensory and cognitive processes (*Ito, 2008*; *Ramnani, 2006*). In this view, cortico-cerebellar interfaces implement essential properties of motor and non-motor (internal) models, that is, representations that can be used to anticipate future events, thereby maximizing the precision of motor, sensory, and cognitive performance (*Ito, 2008*). A particularly salient attribute of such models, under active study, concerns timing. The *cerebellar timing hypothesis* claims that the cerebellum encodes the precise temporal locus of sensory events (*Ivry et al., 2002*; *Spencer and Ivry, 2013*), with potential asymmetric hemispheric sensitivities. While the right cerebellar hemisphere prefers rapid, the left prefers slow signal modulations (*Callan et al., 2007*).

This cerebellar structural and functional organization may support the asymmetric specialization in sampling non-speech (*Boemio et al., 2005*; *Zatorre and Belin, 2001*) and speech sounds (*Poeppel, 2003*) in auditory cortex. Concretely, the sampling of continuous sounds has been argued to proceed in time windows of different lengths, which, in turn, translate into different linguistic segments of speech (e.g., phonemes and syllables) (*Flinker et al., 2019*). Thus, anticipatory modeling of shorter and longer segments may pave the way for optimal sound and speech perception.

If the cerebellum interacts with areas in the cerebral cortex to implement internal models that reflect the temporal structure of perceptual experience, this asymmetry requires a pattern of cross-lateral right-cerebellar-left-cortical and left-cerebellar-right-cortical connectivity. This neurofunctional connectivity pattern is evident in the *motor* domain. Interestingly, similar observations in non-motor domains remain largely unexplored. Is the representation and analysis of temporal information a general feature underpinning cortico-cerebellar functional connectivity? The major goal of this suite of experiments is to fill this gap in our understanding by combining functional and structural prior knowledge with new empirical evidence in a multidimensional approach to systematically contrast fast and slow temporal modulations of sound. The study is anchored in lesion data, which establish the most direct link between function and structure.

Existing deficit lesion data show that damage to the cerebellum impairs the perception of temporal voicing contrasts (*Ackermann et al., 1997*), duration judgments of intervals (*Ivry and Keele, 1989*), and the ability to use temporal event structure to update a representation of the auditory environment (*Kotz et al., 2014*). Similarly, left temporal cortex lesions, in particular, lead to impairments of temporal order judgments (*Efron, 1963*; *Swisher and Hirsh, 1972*), discrimination of rapidly presented complex tone pairs (micropatterns; *Chedru et al., 1978*), phonological discrimination associated with increased detection thresholds for rapid (but not slow) modulations (*Robson et al., 2013*), detection of short timescale voicing contrasts, and increased temporal order thresholds (*Fink et al., 2006*). Together these findings confirm the functional relevance of differential temporal sensitivities in *both* cerebellar and temporal cortex. This evidence then motivates the question of how cerebellum and temporal cortex interface to optimize the processing of spectro-temporal information at different timescales in audition (*Boemio et al., 2005*; *Kotz and Schwartze, 2010*; *Poeppel, 2003*).

Here, we combine lesion mapping, tractography in healthy participants, and behavioral data to gain new mechanistic insight. First, patients, who suffered from a circumscribed stroke in the left posterior superior temporal sulcus (pSTS – with spared Heschl's gyrus, that is, putative primary auditory cortex – are characterized. Based on an extensive literature, such patients are expected to show impaired temporal discrimination for non-verbal and verbal information, restricted to fast modulations, such as voicing and place of articulation contrasts *Boemio et al., 2005*; *Elangovan and Stuart, 2008*; *Rosen, 1992*). We test this hypothesis and aim to replicate prior results, using a range of speech and non-speech materials. Second, we predict that specific lesion-symptom mapping will identify a critical seed region to distinguish the well-documented dorsal and ventral fiber tracts of the temporo-frontal speech network (*Frey et al., 2008*; *Friederici, 2011*; *Saur et al., 2008*; *Turken and Dronkers, 2011*). Third, and most critically, if the generalized timing conjecture is on the right track, anatomic tractrography should reveal direct connections linking the left posterior temporal cortex with the right posterior lateral cerebellum (cerebellar crura I/II), ostensibly engaged in auditory processing (*Petacchi et al., 2005*). Assessing connectivity in healthy participants based on lesion information is a relatively new method that measures structural disconnection in networks associated with given anatomical regions (*Foulon et al., 2018*). This allows for the indirect estimation of the lesion effect on structural brain networks. In this regard, it was shown that behavioral deficits can be explained similarly by local brain damage and indirectly measured disconnection (*Salvalaggio et al., 2020*).

Modeling and adapting to a dynamic auditory environment require a sufficiently detailed representation of the spectro-temporal structure of sound. Internal models of these sound properties must play an essential role in optimizing proactive perceptual and cognitive performance. Speech, as a particularly complex sound signal, evolves over different timescales and requires spectral and temporal segmentation in establishing building blocks for the construction of models of the auditory world. This fundamental task likely relies on the precise orchestration of cortical and subcortical brain areas (*Kotz et al., 2014*; *Schwartze and Kotz, 2016*). An integrative theoretical interpretation of the predicted results from the perspective of a cerebellar-temporal cortex interface – with potential lateralization reflecting differential temporal sensitivities – offers an intriguing new perspective to explore the anatomical basis of cerebellar internal modeling in motor control and audition, providing a computational generalization that may offer useful new angles for experimentation.

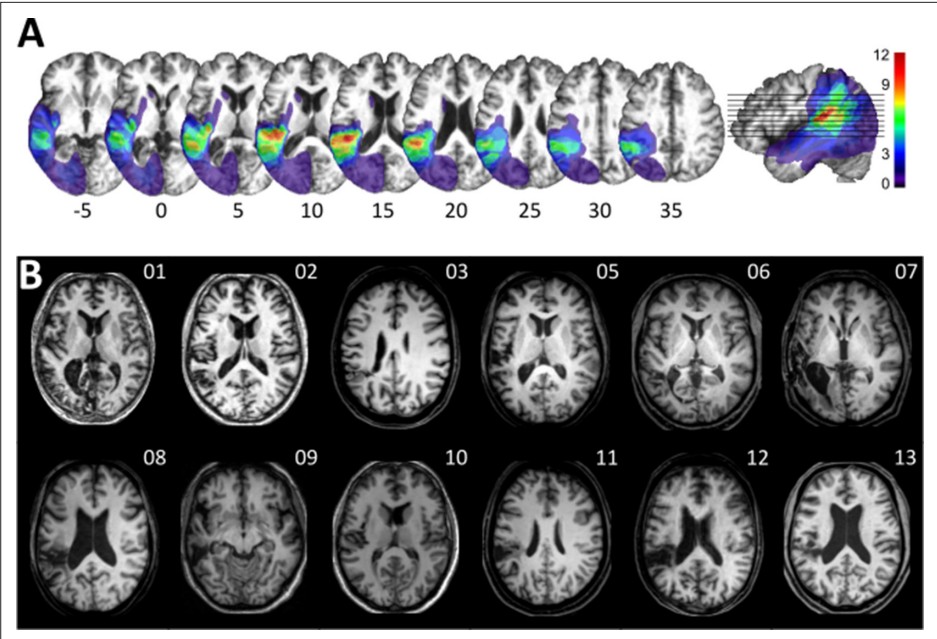

**Figure 1.** Visualization of lesion distribution. (**A**) (Top row) Lesion frequency map: lesion distribution in the 12 patients superimposed on the scalp-stripped mean patient T1-weighted image. Colorbar specifies the number of patients with overlapping lesions in each voxel, with hot colors indicating that a greater number of patients had lesions in this region. Maximum lesion overlap in left posterior superior temporal gyrus (planum temporale) and underlying white matter (MNI −45,−36, 15). (**B**) (Bottom row) MRI imaging showing lesion location on a representative axial slice.

## Results

### Lesions and behavioral deficits

Twelve patients with chronic left temporal stroke and 12 matched controls (*Supplementary file 1*) were mapped and tested using auditory temporal order and discrimination tasks (same-different judgments). We tested the groups on a range of perceptual tasks selected to probe temporal processing in hearing. *Figures 1 and 2* depict the lesion distribution and task performance, respectively.

Tonal stimuli were presented to determine individual threshold levels for temporal order judgment and micropattern discrimination, which were then linked to normal performance in controls (*Figure 2*). These tasks index a participant's perceptual abilities to (i) encode non-verbal short timescale spectral information and (ii) compare a current stimulus to the representation of a preceding stimulus. The term 'micropattern' describes pairs of complex tones presented with stimulus onset asynchronies (SOAs) below an individual's temporal order threshold, that is, the shortest SOA at which the temporal order of two tones can be perceived (*Chedru et al., 1978*; *Anstis et al., 1978*). In this case, the order of still discriminable different stimulus elements cannot be determined. However, frequency reversals within a micropattern lead to a perceptual dissociation. Micropatterns are perceived as relatively lower or higher in pitch, a phenomenon attributed to the perceptual dominance of the second stimulus frequency (*Efron, 1973*).

Same-different judgments were used to assess the discrimination performance for minimal-pair words (e.g., Dach [dax] (engl. *roof*) – Bach [bax] (engl. *stream*)), non-words (e.g., Pach [pax] – Kach [kax]), and phonemes (e.g., /tr/ and /pr/) differing in contrastive phonological features. Corresponding error rates served to compare basic perceptual speech abilities for the encoding and modeling of different levels of speech and contrastive features (e.g., articulation, voicing, manner of articulation) relative to controls (*Figure 2D*). See Materials and methods for details.

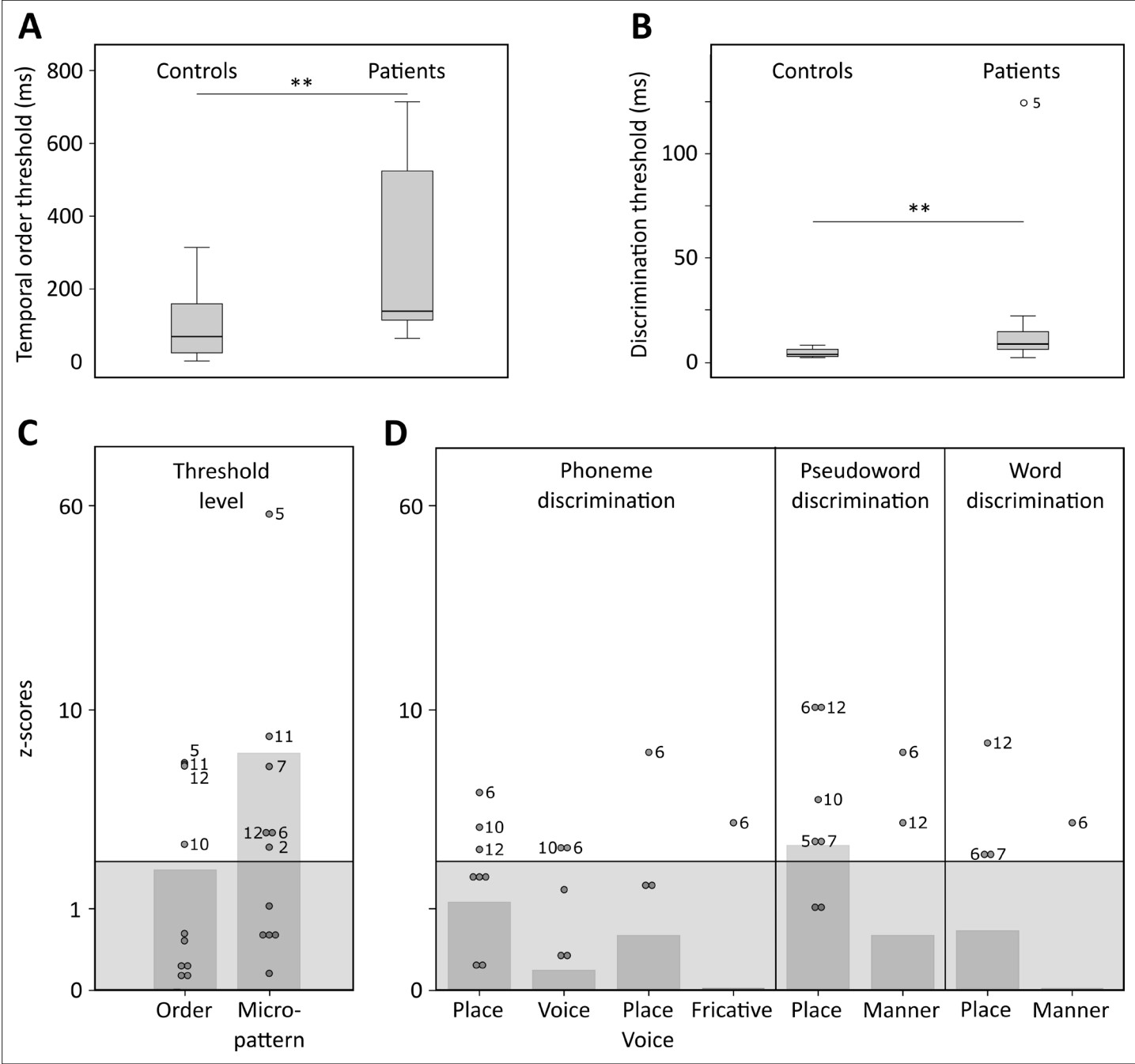

**Figure 2.** Temporal order and discrimination thresholds and identification of deficit-positive (LG+) and -negative (LG−) lesion group. Boxplots display median (horizontal line), first and third quartile (box), data range (whiskers), and outlier (dot) of the threshold levels in milliseconds for temporal order judgments (A) and discrimination of micropatterns (B) in the control and patient group. Patients as compared to controls show higher temporal order and micropattern discrimination thresholds. To identify deficit-positive (LG+) and -negative (LG−) lesion groups, patients' mean (bars, dark gray) and individual performance (circles) on temporal order and micropattern discrimination (C) and phoneme/word discrimination (D) were converted to into z-scores relative to control group means for each behavioral test. Values > 0 indicate worse performance than controls within (light gray) and outside (no color) plus two standard deviations (SD) of the controls mean. Patients scoring outside two SD of the controls (impaired performance, LG+) are indicated by subject number.

The online version of this article includes the following source data for figure 2:

**Source data 1.** Behavioral data.

**Table 1.** Between-group comparisons of error rates for verbal discrimination tasks per category and contrastive feature.

| Variable | Group | | Test statistics | | |
|---|---|---|---|---|---|
| *Category* | *Patients (N = 12)* | *Controls (N = 12)* | *Category effect* | | *Group effects* |
| | | | *Patients* | *Controls* | |
| Words (W) | 0.02 ± 0.02 | 0.01 ± 0.01 | $\chi^2(2)$ = 10.86 p = 0.003 | $\chi^2(2)$ = 5.72 p = 0.056 | U = 83, p = 0.276 |
| Pseudowords (PW) | 0.05 ± 0.05 | 0.02 ± 0.03 | | | U = 96, p = 0.089 |
| Phonemes (P) | 0.07 ± 0.09 | 0.04 ± 0.07 | | | U = 95, p = 0.099 |
| *Feature* | *Patients (N = 12)* | *Controls (N = 12)* | *Feature effects* | | *Group effects* |
| | | | *Patients* | *Controls* | |
| Place (W) | 0.04 ± 0.08 | 0.01 ± 0.03 | Z = –1.34 p = 0.250 | Z = –1 p = 0.500 | U = 79, p = 0.366 |
| Manner (W) | 0.01 ± 0.02 | 0.01 ± 0.02 | | | U = 72, p = 0.500 |
| Place (PW) | 0.16 ± 0.22 | 0.03 ± 0.05 | Z = –2.38 p = 0.008 | Z = –1.342 p = 0.250 | U = 101.5, p = 0.045 |
| Manner (PW) | 0.02 ± 0.05 | 0.01 ± 0.02 | | | U = 78.5, p = 0.366 |
| Place (P) | 0.23 ± 0.23 | 0.08 ± 0.14 | $\chi^2(2)$ = 18.31 p = 0.00004 | $\chi^2(2)$ = 7.0 p = 0.065 | U = 99.5, p = 0.057 |
| Fricatives (P) | 0.03 ± 0.12 | 0.03 ± 0.12 | | | U = 72, p = 0.500 |
| Voice (P) | 0.07 ± 0.1 | 0.06 ± 0.08 | | | U = 70.5, p = 0.466 |
| Place and voice (P) | 0.05 ± 0.15 | 0.03 ± 0.08 | | | U = 72.5, p = 0.500 |

Mean relative error rates (± SD) and non-parametric test statistics for within- (Friedman $\chi^2$ and Wilcoxon signed-rank Z-statistics, Bonferroni-adjusted significance levels set at p < 0.017) and between-subject comparisons (Mann-Whitney U test, exact p-values [one-sided]) for each category and contrastive feature.

## Impaired encoding of non-verbal and verbal spectro-temporal information

Healthy controls showed typical threshold values comparable to previously reported temporal order (SOAs above 15–60 ms) and micropattern discrimination tasks (SOAs of at least of 5 ms) (*Efron, 1963*; *Efron, 1973*; *Hirsh and Sherrick, 1961*; *Yund and Efron, 1974*). In contrast, patients required longer intervals to judge the order of two different frequency tones (U = 114, p = 0.007, effect size *r* = 0.495; *Figure 2A* and *Supplementary file 1*) and to discriminate two micropatterns (U = 120.5, p = 0.002, *r* = 0.572; *Figure 2B* and *Supplementary file 1*). Order and discrimination thresholds were positively correlated in patients (p = 0.015, Spearman's rho *rs* = 0.681) and controls (p = 0.006, *rs* = 0.735). Error rates differed in patients and controls for the discrimination of words, pseudowords, and phonemes (Friedman test, $\chi^2(2)$ = 16.026, p = 0.0002). Phoneme discrimination displayed increased error rates compared to words (Z = –3.624, p = 0.0003, *r* = 0.740, Bonferroni adjustment at p < 0.017) and pseudowords compared to words (Z = –2.684, p = 0.007, *r* = 0.548). Although healthy controls did not show any category effect, patients ($\chi^2(2)$ = 10.857, p = 0.003) showed higher error rates for phonemes than words (Z = –2.287, p = 0.015, *r* = 0.660), and for pseudowords than words (Z = –2.666, p = 0.002, *r* = 0.769). Subsequent between-group comparisons revealed non-significant trends for higher error rates in patients compared to controls for the discrimination of pseudoword and phoneme pairs (*Table 1*). Thus, in line with previous results (*Chedru et al., 1978*; *Efron, 1963*; *Swisher and Hirsh, 1972*), we confirm that patients with left posterior temporal strokes show less accurate auditory spectro-temporal processing and concomitant perceptual speech deficits.

## Feature specific impairment for place of articulation contrasts

As the left-lateralized lesion patients showed robustly worse pseudoword and phoneme discrimination, we next tested phoneme specific features focusing specifically on the shortest timescales (*Rosen, 1992*). Only patients showed significant effects for contrastive features for phoneme ($\chi^2(2)$ = 18.313, p = 0.00004) and pseudoword pairs (Z = –2.38, p = 0.008, *r* = 0.687). Post hoc Wilcoxon

tests (Bonferroni-adjusted significance level at p < 0.0083) confirmed higher error rates for the discrimination of place of articulation contrasts than for voicing (Z = −2.521, p = 0.004, r = 0.728) in phonemes. The same was true when comparing place of articulation to combined place of articulation and voicing contrast (Z = −2.536, p = 0.004, r = 0.732) as well as for place of articulation compared to fricative contrasts (Z = −2.555, p = 0.004, r = 0.738). Similarly, patients showed higher error rates for place of articulation relative to manner of articulation contrasts (Z = –2.684, p = 0.004, r = 0.775) in pseudowords. The same tests only yielded a non-significant trend for contrastive features within the phoneme category ($\chi^2(2) = 7.0$, p = 0.065) in healthy controls. Subsequent between-group comparisons revealed higher error rates for the discrimination of place of articulation contrasts in pseudowords in patients but not controls (U = 101.5, p = 0.045, r = 0.348) and a non-significant trend (U = 99.5, p = 0.057, r = 0.343) for the discrimination of place of articulation contrasts in phonemes (**Table 1**). In sum, patients displayed speech processing deficits preeminent for information encoded in the spectro-temporal fine structure at short timescales (**Rosen, 1992**).

## Linking auditory spectro-temporal and short timescale phonemic processing

Threshold levels for temporal order and micropattern discrimination were correlated (Spearman's rank-order correlations, one-sided) with performance measures for different discriminative features to assess possible associations between impaired auditory temporal processing and the processing of phonemic cues encoded at short timescales. There was a positive correlation between error rates for voicing contrasts in phoneme discrimination with threshold levels for auditory order (p = 0.032, r = 0.549) and micropattern discrimination (p = 0.043, r = 0.516). The associations between increased error rates for place of articulation contrasts with increased auditory order (p = 0.051, r = 0.495) and discrimination thresholds did not meet conventional significance (p = 0.061, r = 0.471).

Importantly, no significant associations were found between lesion volume or hearing loss and threshold values or error rates for feature discrimination (**Supplementary file 1**). In line with our hypotheses and previous findings (**Fink et al., 2006**; **Robson et al., 2013**), we show that *patients exhibit a short timescale specific perceptual deficit* for tones and phonemes.

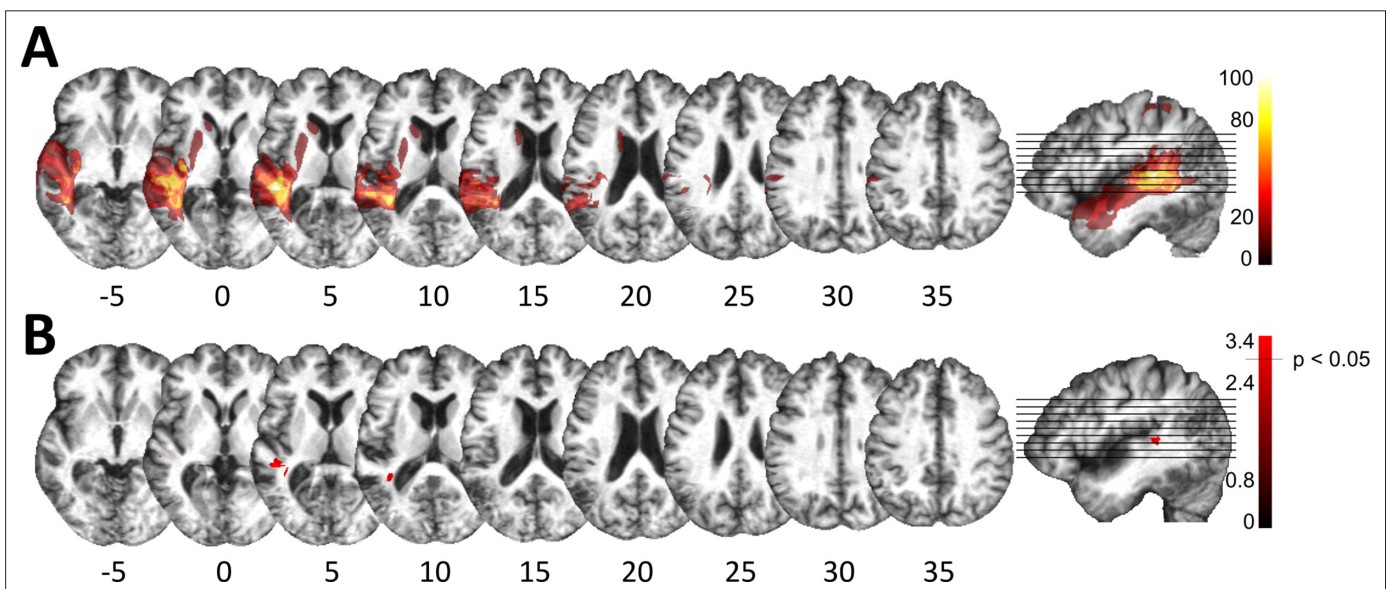

**Figure 3.** Lesion analysis of deficit-positive (LG$^+$) and -negative (LG$^−$) lesion group. (**A**) Subtraction plot shows voxels more frequently damaged in LG$^+$. Colorbar specifies relative frequency (percentage) of overlapping lesions in the patient group with impaired performance (LG$^+$) after subtracting lesion overlap of LG$^−$ from lesion overlap of LG$^+$. (**B**) Voxelwise statistical analyses (Liebermeister measure for binomial data, permutation FWE-corrected z-scores at α-level of p < 0.05): lesions in posterior superior temporal sulcus (STS) (Montreal Neurological Institute [MNI] −48, −34, 5 and −38, −43, 10) are significantly associated with impaired temporal information processing (LG$^+$).

The online version of this article includes the following figure supplement(s) for figure 3:

**Figure supplement 1.** Definition of control regions for tractography.

## Posterior superior temporal injury is associated with impaired spectro-temporal encoding

Despite consistent group-level effects, patients varied in terms of their encoding capacity for tones and phonemes. We therefore next explored whether performance variability links to lesion sites within the left temporal cortex. Subsequent analyses related deficient spectro-temporal processing at short timescales to specific temporal lesion sites (lesion-symptom mapping). Performance differences between healthy controls and patients with or without a performance impairment (see *Figure 2*) allowed identification of brain regions more frequently associated with an impairment (*Figure 3*; see Materials and methods for more details). This procedure effectively split the patient group into two sub-groups of equal size, as six patients showed impaired performance in at least two subtests (deficit-positive lesion group, LG⁺) as compared to six patients (deficit-negative lesion group, LG⁻) performing similarly to controls (*Figure 2C–D*). The patient groups did not differ in terms of demographic or clinical characteristics (*Supplementary file 1*). Impairments in other cognitive domains (attention, memory, executive function) were present in some patients, but not exclusively in those who belonged to LG⁺ (*Supplementary file 1*). The maximal lesion overlap (*Figure 1A*) was centered on the posterior superior surface of the left superior temporal gyrus (STG) (planum temporale) extending into the underlying

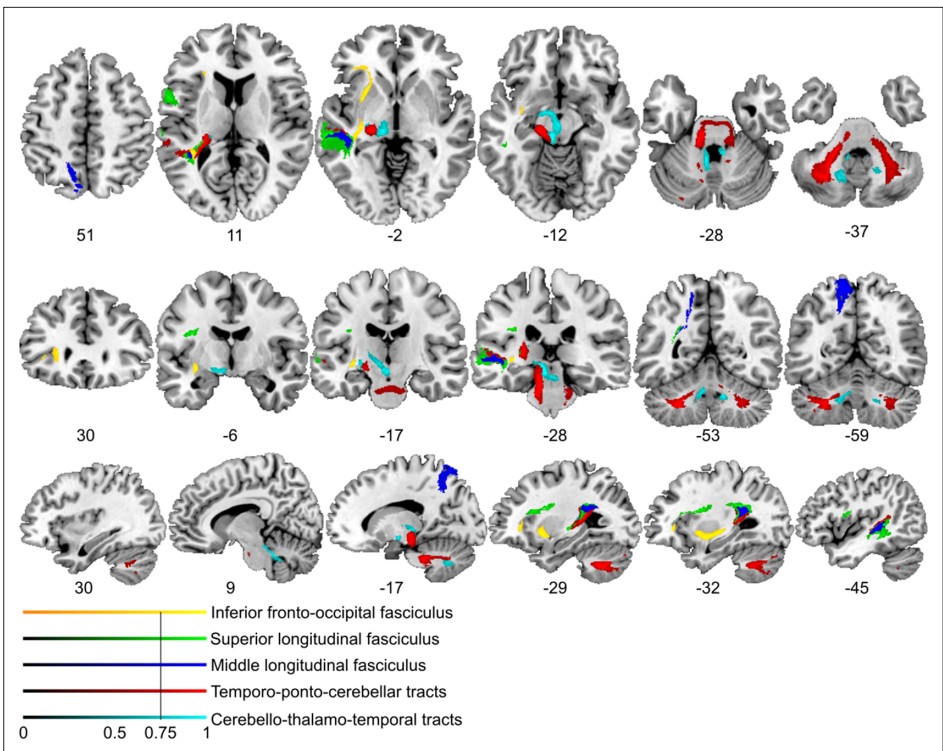

**Figure 4.** Lesion-informed probabilistic tractography. Diffusion tractography based on a dataset of 12 healthy controls. Seed areas only included voxels being more frequently associated with impaired processing of temporal information. Inclusion masks were used to subdivide individual connectivity distributions into separate fiber bundles. The tracts are superimposed on the MRIcron ch2bet template in standard Montreal Neurological Institute (MNI) space (axial, coronal, and sagittal slices, corresponding MNI coordinates are indicated below). Displayed group variability maps result from binarized tract volumes (thresholded connectivity distributions) that quantify the percentage of subjects (>75%) showing connectivity between the seed masks and the respective voxel (values range from 0.0 to 1.0). Yellow: inferior fronto-occipital fasciculus (IFOF), green: superior longitudinal fasciculus (SLF), red: temporo-ponto-cerebellar tracts, dark blue: middle longitudinal fasciculus, light blue: cerebello-rubro-thalamic tract.

The online version of this article includes the following figure supplement(s) for figure 4:

**Figure supplement 1.** Comparison of cortico-cortical connectivity from deficit-negative control region.

**Figure supplement 2.** Comparison of cortico-cerebellar connectivity from control region in M1.

**Figure supplement 3.** Comparison of cerebello-cortical connectivity from control region in M1.

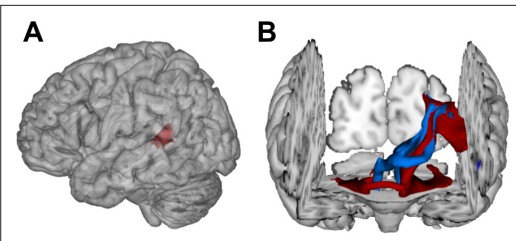

**Figure 5.** Visualization of temporal cortex-cerebellum connectivity. Bilateral and bidirectional connectivity of seed regions (**A**) in the left posterior superior temporal sulcus (pSTS). (**B**) Temporo-ponto-cerebellar tracts (red) and cerebello-rubro-thalamo-temporal tracts (blue) connect pSTS with the postero-lateral cerebellum and the dentate nucleus, respectively.

white matter. In contrast, the lesion subtraction of LG⁻ patients from LG⁺ patients (*Figure 3A*) linked pSTS, adjacent middle temporal gyrus (MTG), and white matter below the STS to the spectro-temporal encoding impairment in the LG⁺ group. Differences in lesion distribution between LG⁻ and LG⁺patients were statistically confirmed (Liebermeister test, binomial data with 1 = LG⁺ and 0 = LG⁻, permutation FWE-corrected $\alpha$-level of p < 0.05) (*Figure 3B*).

## Posterior superior temporal projections interface spectro-temporal processing networks with the cerebellum

We next used the respective areas as seed regions for probabilistic fiber tractography in a healthy age-matched sample to visualize the underlying common connectivity pattern (see Materials and methods). Thus, we indirectly explored the association between posterior superior temporal discon-nection and processing of sound at short timescales. Long association and projection fibers origi-nating from the seed masks in the pSTS and the subjacent white matter below the STS (*Figure 3B*) were identified within the anterior floor of the left external capsule, the left periventricular white matter, and the brain stem. Subdivision into separate fiber bundles (*Figure 4*) indicated connectivity along the inferior fronto-occipital fasciculus (IFOF) traveling within the anterior floor of the external capsule. Terminations were present in the left inferior frontal gyrus (pars triangularis, Brodmann area [BA] 45) and in the lateral orbitofrontal cortex (BA 47). Association fibers also extended along the posterior lateral surface of the lateral ventricle to the left superior parietal cortex (BA 7). Considering the characteristic connectivity patterns of these association fibers with the superior parietal cortex, they may correspond to the posterior middle longitudinal fasciculus (MLF) (*Makris et al., 2013a*; *Wang et al., 2013*). Other cortico-cortical association fibers curved upward and traveled rostrally within the periventricular white matter lateral to the corona radiata in the superior longitudinal fascic-ulus (SLF) with terminations in the left inferior frontal gyrus (pars opercularis, BA 44) and the left dorsolateral prefrontal cortex (BA 9, BA 46).

Most importantly, fibers originating in the superior posterior temporal cortex (*Figure 5A*) reached the posterolateral cerebellum (crura I and II) and the dentate nuclei bilaterally (*Figure 4* and *Figure 5B*). These fibers coursed rostrally and ascended medially near the posterior temporal and inferior parietal cortex before they descended through the retrolenticular internal capsule along the left cerebral peduncle to the pontine nuclei and the ipsilateral middle cerebellar peduncle (MCP). They additionally decussated at the ventral pons to the right cerebellar peduncle, giving rise to bilateral temporo-ponto-cerebellar tracts (*Figure 5B*). Other fibers conformed to the cerebello-rubro-thalamic tract connecting the posterior superior temporal cortex with the bilateral dentate nuclei along the superior cerebellar peduncles (SCPs). They crossed at the level of the inferior olive to the left red nucleus and projected to the posterior thalamus (pulvinar) (*Figures 4 and 5B*). Although the cerebello-rubro-thalamic tract is considered a decussating pathway, there is some evidence for a non-decussating pathway and for specific connections of these pathways to more anterior and lateral as opposed to more posterior and medial thalamic targets (*Petersen et al., 2018*). Considering that the differential connectivity of Broca's area with the thalamus includes the pulvinar, one may speculate that the non-decussating pathway also supports language function (*Bohsali et al., 2015*).

Taken together, the results demonstrate that lesions in the left pSTS led to a spectro-temporal processing deficit at short timescales for lower-level auditory and speech information. The neural substrate for such perceptual necessities is revealed here for the first time: a bidirectional temporo-cerebellar connectivity confirmed by probabilistic tractography.

## Control analysis

To test for deficit specificity of the identified temporo-cerebellar networks, different regions (IPL, AG, pMTG) taken from the same study population served as negative control seed regions (*Figure 3— figure supplement 1B*). To further test for the specificity of cerebellar terminations, we chose another independent control region in the motor cortex approximately corresponding to the left foot motor area. The respective areas served as seed regions for probabilistic fiber tractography in a healthy age-matched sample to visualize the underlying common connectivity pattern (see Materials and methods – Control analysis). Fibers originating from the seed masks in the control region (left IPL, AG, pMTG) were identified in periventricular white matter. Using the same waypoint mask in the left periventricular white matter lateral to the superior corona radiata, we found connectivity along the SLF (*Figure 4—figure supplement 1*). Unlike fibers that originate in the STS, the bundle traveled further cranially with terminations in the left supplementary motor cortex corresponding to BA 6. Based on its anatomical characteristics, these fibers likely belong to the second branch of the SLF II. This demonstrates that probabilistic tractography from a nearby control region that uses the same waypoint masks can in principle separate different fiber bundles. Tractography from these control seed regions however showed no relevant cortico-cerebellar connectivity along the MCP and SCP. This supports the specificity of the reported temporo-cerebellar tracts originating from the STS in the context of impaired processing of sound at short timescales. Tractography from the second control seed region in the foot area of the motor cortex revealed bilateral connectivity along the pyramidal tract with terminations in cerebellar lobulus VIII (*Figure 4—figure supplement 2*) that corresponds to regions associated with motor processing (i.e., the motor cerebellum) (*Stoodley et al., 2012*). The ascending cerebellar tracts along the SCP were not separable at the level of the cerebellum, which is likely due to the low resolution of the method and close proximity of fibers in the dentate nucleus and SCP. In contrast, we found different projections in the thalamus, where fibers to the motor cortex could be delineated in the ventral lateral thalamic nuclei and fibers to the temporal cortex in the posterior thalamus (*Figure 4—figure supplement 3*).

These results further demonstrate the specificity of temporo-cerebellar and thalamo-temporal projections in relation to impaired processing of sound at short timescales.

## Discussion

Our functional anatomic discovery, deriving from deficit lesion data from patients as well as tractography data from healthy participants, provides a structural basis for a mechanistic link between two domains of inquiry that have proceeded largely independently: research on the temporal dynamics of auditory perception and research on internal models supported by the cerebellum. The newly described cortico-cerebellar connectivity forms the basis for how the specific anatomic layout underpins one key aspect of auditory perceptual analysis. We set out to answer the following questions: First, how does the asymmetrical specialization in sampling of verbal and non-verbal sound information at different timescales tie in with the encoding of spectro-temporal structure in an internal model framework? Second, do we have to consider cross-lateral cortico-subcortical structural connectivity to achieve a comprehensive view of asymmetrical sampling of sound properties? Lesion-symptom-informed probabilistic tractography, seeded in the left pSTS of healthy participants, revealed temporo-frontal and bidirectional structural connectivity with the cerebellar dentate nuclei and crura I/II (see also *Sokolov et al., 2014* for right pSTS connectivity). The evidence we describe (i) shows that lesion-related deficits in spectro-temporal analysis occur in posterior temporal regions connected to the cerebellum and (ii) is in line with the concept of a generalizable role of cerebellar-mediated internal models that extends beyond motor control to auditory perception.

### Impaired auditory spectro-temporal encoding for non-verbal and verbal information

The tested patient sample (see *Figure 1*) displayed only mild aphasic symptoms (*Supplementary file 1*) but higher temporal order thresholds (*Figure 2A*), falling into the range of 150–600 ms relative to 15–60 ms previously reported for healthy participants (*Efron, 1963*; *Fink et al., 2006*). Higher micro-pattern discrimination thresholds in patients (*Figure 2B*) converge with evidence for impaired micro-pattern discrimination in patients with selective temporal compared to frontal lesions (*Chedru et al.,*

*1978*). In patients, discrimination of the place of articulation contrast was impaired for phonemes and pseudowords but not words (*Figure 2D*). This feature is likely represented in the spectro-temporal fine structure occurring over short timescales (20–50 ms), whereas phonemic contrasts to voicing and manner are encoded in timescale duration differences of voice-onset times (*Elangovan and Stuart, 2008*) and the slowly varying temporal envelope (50–500 ms; *Rosen, 1992*). The redundancy of information across shorter and longer timescales for the latter phenomena may contribute to this feature specific effect. Both phoneme discrimination and auditory order or micropattern discrimination are assumed to map onto lower-level auditory processes on short timescales. In contrast to earlier findings (*Chedru et al., 1978*), we show that discrimination thresholds and discrimination performance for non-verbal and verbal information go hand in hand. This suggests a common process that contributes to short timescale spectro-temporal encoding for both non-verbal *and* verbal information – and that impaired auditory temporal processing can (at least partly) explain lower-level auditory deficits cascading into speech comprehension deficits (*Robson et al., 2013*).

## Posterior superior temporal regions link cortico-cortical and subcortico-cortical spectro-temporal processing networks

Lesion-symptom mapping (*Figure 3*) identified an area in the pSTS that was more frequently affected in patients with impaired short timescale spectro-temporal encoding for non-verbal and verbal information. This is in line with the dissociation of information unfolding over timescales corresponding to global prosodic, syllabic, and phonemic levels that is mirrored by a potential functional asymmetric temporal sensitivity of higher-order auditory areas in the superior temporal cortices. These are thought to preferentially encode rapidly changing auditory signals in time windows of ~20–40 ms in the left hemisphere and of ~150–250 ms in the right hemisphere (*Boemio et al., 2005*; *Flinker et al., 2019*; *Poeppel, 2003*).

Our tractography findings of *cortico-cortical structural connectivity* are in line with results mapping speech processing networks that connect to the left MTG/STS along the IFOF and the SLF with BA 47 and BA 46 (*Turken and Dronkers, 2011*). Functionally distinct subdivisions of the SLF, namely the SLF III and arcuate fasciculus, may provide higher-order (somato-)sensory and auditory input to inferior frontal (BA 44) or dorsolateral prefrontal regions (BA 46, BA 6/8; *Makris et al., 2005*). Other studies revealed connectivity along the middle longitudinal fascicle between the posterior temporal cortex and the superior parietal lobe (BA 7; *Makris et al., 2013b*; *Wang et al., 2013*), an area associated with audio-visual multisensory integration (*Molholm et al., 2006*).

We add a novel contribution to this established pattern of cortico-cortical connectivity by revealing cross-lateral and ipsilateral structural cortico-subcortical connectivity, with clear implications for the emerging cerebellar contributions to higher cognitive functions. To date, such contributions have been related to reciprocal prefrontal-cerebellar and posterior parieto-cerebellar projections, connecting association cortices with the lateral and posterior cerebellar hemispheres (*Brodal, 1978b*; *Brodal, 1978a*; *Brodal, 1979*; *Jissendi et al., 2008*; *Kelly and Strick, 2003*; *Ramnani et al., 2006*). Earlier claims considering temporo-cerebellar projections based on diffusion-weighted imaging of the cerebral peduncles as insignificant in both humans and non-human primates (*Ramnani et al., 2006*) should be revised, as the topographical distribution of cortico-ponto-cerebellar projections likely extends beyond and is distinct from the most prominent prefrontal and primary motor structural connectivity patterns (see Results – Control analysis and *Figure 4—figure supplement 3*). As demonstrated here, the cortico-cerebellar system may also comprise reciprocal ipsi- and contralateral temporo-cerebellar projections that so far have gained only very little attention (*Schmahmann and Pandya, 2009*; *Schmahmann and Pandya, 1991*; *Sokolov et al., 2014*). Moreover, the present findings confirm fibers connecting the superior posterior temporal cortex and the posterior lateral cerebellum (crura I/II, cerebello-rubro-thalamic tract) and indicate fibers originating in bilateral dentate nuclei, which run along the SCP and posterior thalamus (cerebello-rubro-thalamic tract). These observations confirm the concept of reciprocal cortico-cerebello-cortical loops (*Salmi et al., 2010*) and conform to anatomical landmarks for cerebro-cerebellar connections in the brain stem and thalamus demonstrated in previous neuroanatomical and MRI studies (*Bernard et al., 2014*; *Brodal, 1978b*; *Brodal, 1979*; *Dum and Strick, 2003*; *Habas and Cabanis, 2006*; *Habas and Cabanis, 2007*; *Pandya et al., 1994*; *Schmahmann and Pandya, 1991*).

Consequently, the need arises to integrate these novel structural connectivity findings in a theoretical way (i) with well-established functional evidence for spectro-temporal sound processing at different time scales in temporal cortices (*Boemio et al., 2005*; *Callan et al., 2007*; *Poeppel, 2003*) and (ii) with the cerebellum's critical role in supporting internal models (*Kotz and Schwartze, 2010*; *Schwartze et al., 2012*).

## The temporo-cerebellar interface is linked to spectro-temporal encoding

In contrast to the bilateral posterior lateral cerebellar contributions to auditory (*Pastor et al., 2002*; *Petacchi et al., 2005*) and temporal processing (*Ivry et al., 2002*; *Keele and Ivry, 1990*; *Spencer and Ivry, 2013*), there is sparse evidence on temporo-cerebellar coupling (*Pastor et al., 2002*; *Pastor et al., 2006b*; *Pastor et al., 2008*). However, such coupling may support the continuous updating of internal models of the auditory environment (*Kotz et al., 2014*) based on precise encoding of temporal structure. For example, functional imaging studies show increased induced oscillatory activity in response to 40 Hz auditory stimulation (corresponding to a sampling period of ~25 ms) in the bilateral posterolateral cerebellum (crus II) (*Pastor et al., 2002*), and effective connectivity between superior temporal areas (STG and STS) and cerebellar crus II increases with 40 Hz auditory stimulation (*Pastor et al., 2008*). Likewise, cortical oscillatory responses to 40 Hz auditory stimulation

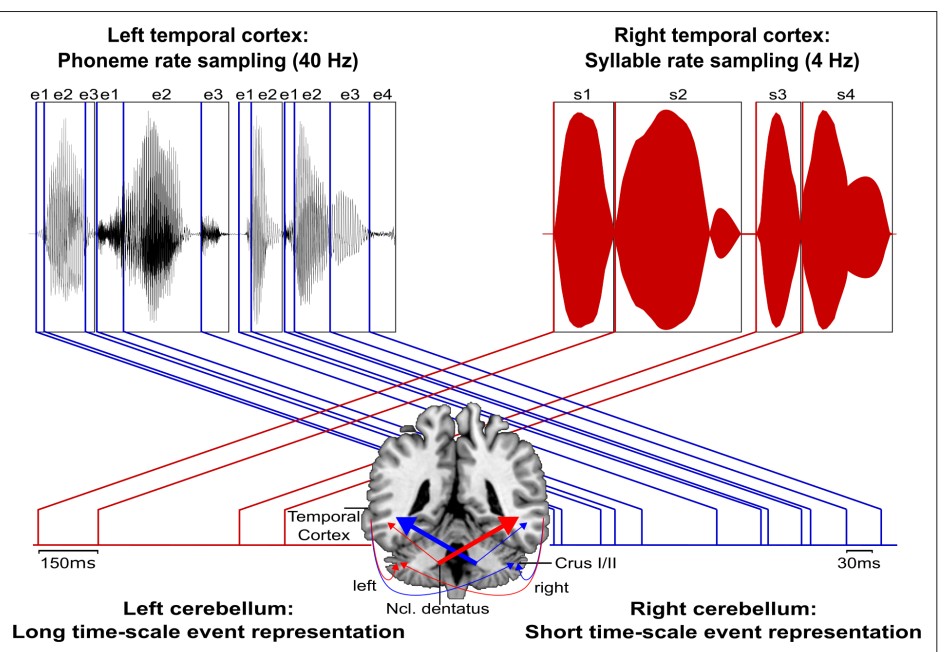

**Figure 6.** Schematic conceptualization of temporo-cerebellar interaction for internal model construction in audition. Differential temporo-cerebellar interaction model depicting hypothesized connectivity between areas in the temporal lobe and cerebellum that may underlie sound processing at different timescales. Left and right cerebellum contribute to the encoding of event boundaries across long (red) and short (blue) timescales, respectively (*Callan et al., 2007*). These event representations are extracted from salient modulations of sound properties, that is, changes in the speech envelope (fluctuations in overall amplitude, red) corresponding to syllables (**s1–s4**) and the fine structure (formant frequency transitions, blue) corresponding to phonemes (**e1–e4**) (*Rosen, 1992*; *Weise et al., 2012*). Reciprocal ipsi- and cross-lateral temporo-cerebellar interactions between temporal cortex, crura I/II, and dentate nuclei yield unitary temporally structured stimulus representations conveyed by temporo-ponto-cerebellar and cerebello-rubro-thalamo-temporal projections (arrows). The resulting internal representation of the temporal structure of sound sequences, for example, speech, fits the detailed cortical representation of the auditory input to relevant points in time to guide the segmentation of a continuous input signal (waveform) into smaller perceptual units (boxes). This segmentation is further guided through weighting of information (symbolized by arrow thickness) towards the short and long timescale of sound processing in the left and right temporal cortex, respectively. This process allows distinctive sound features (e.g., word initial plosives /d/ (e1 in s1), /t/ (e1 in s2), and /b/ (e1 in s3) varying in voicing or place of articulation) to be optimally integrated at the time of their occurrence.

diminish after inhibitory cerebellar transcranial magnetic stimulation (*Pastor et al., 2006b*). Moreover, induced gamma band oscillatory activity in the auditory cortex as well as in the cerebellum in response to random auditory stimulation tightly time-locks to auditory stimulus onsets (*Demiralp and Basar-Eroglu, 1996*).

The cerebellum likely supports the encoding of an event-based representation of the temporal structure extracted from the auditory input signal by tracking salient modulations (e.g., changing sound energy levels as in onset transients or frequency transitions) of the physical sound across time that contribute to the segmentation of a continuous auditory input signal into smaller perceptual units. Such reciprocal temporo-cerebellar interactions might provide a unitary stimulus representation at different timescales for later processing stages (*Schwartze and Kotz, 2016*; *Schwartze et al., 2012*; *Weise et al., 2012*). The overlap with the encoding of auditory signals in temporal integration windows of different lengths (*Boemio et al., 2005*) and possible cerebellar encoding of event boundaries across different timescales (*Callan et al., 2007*) suggests that a temporally structured event representation may map the detailed cortical representation of auditory sensory information to relevant points in time (*Figure 6*), allowing for salient sound features to be optimally processed (*Kotz and Schwartze, 2010*; *Schwartze and Kotz, 2013*; *Schwartze and Kotz, 2016*; *Schwartze et al., 2012*). Contrary to our original hypothesis, we found both ipsi- and cross-lateral cortico-cerebellar connectivity. In this regard, *Boemio et al., 2005* showed in an fMRI study that the STS but not the STG shows duration-sensitive lateralization for shorter and longer timescales. Based on these findings the authors proposed that the bilateral STS receive input differently from the STG through intra- and interhemispheric fibers, weighting information toward short timescales of sound processing. Such weighting might be guided by event boundaries across different timescales encoded in both cerebellar hemispheres.

Notably, in auditory processing the prefrontal cortex (supplementary motor area, BA 6; *Pastor et al., 2006a*) and its connections to the cerebellum (*Aso et al., 2010*; *Jissendi et al., 2008*) play an essential role in the temporal integration of sensory information (*Schwartze and Kotz, 2013*; *Schwartze et al., 2012*). This extended network, consisting of prefrontal cortex, temporal cortex, and cerebellum, provides a platform to integrate sensory input over different timescales to continuously update spectro-temporal models of the auditory environment, thus optimizing sound processing (*Kotz and Schwartze, 2010*; *Kotz et al., 2014*; *Schwartze and Kotz, 2013*; *Schwartze et al., 2012*).

Many investigations have put forward theories on the cerebellum's role in perception and cognition (for a review, see *Baumann et al., 2015*), and the illustrated cortico-subcortical and cortico-cortical structural connectivity pattern is presumably not unique for auditory temporal processing but showcases one possible general role for perceptual processing.

## Limitations

This study has potential limitations. First, the study population is relatively small and lesion-symptom mapping is typically applied to larger populations with wider lesion distribution. Although careful selection of circumscribed lesions has the advantage of highlighting behavioral differences without confounding other deficits (e.g., primary auditory processing), it is possible that additional regions are involved in processing of sound at short timescales. However, tractography based on healthy participants makes it possible to indirectly obtain information (i.e., structural disconnection) about brain regions contributing to the investigated function. In addition, it is likely that the small number of patients might hamper the ability to detect statistically significant differences between the behavior of controls and patients. Nevertheless, we are confident that the current results align with the fact that the posterior superior temporal cortex contributes to the processing of sound at short timescales, as indicated by previous neuropsychological evidence and lesion studies (*Boemio et al., 2005*; *Chedru et al., 1978*; *Efron, 1963*; *Robson et al., 2013*; *Swisher and Hirsh, 1972*). Further studies should however test larger populations to replicate and extend this finding.

Second, although the results confirm the processing of sound at short timescales in the left STS in patients with left temporal lesions, we did not include a right hemisphere patient control group to test for lateralization. This would be problematic in the first place as right hemisphere lesions tend to be more extensive and rarely spare the primary auditory cortex. While a comparison of left and right temporal lesions would have allowed distinguishing processing differences of shorter and longer timescales, such a comparison would have been likely confounded by a primary auditory processing

deficit. Future studies could overcome this problem by using a virtual lesion approach (i.e., by applying inhibitory transcranial magnetic stimulation) that would allow for reversible deactivation of left and right pSTS to test for verbal and non-verbal processing differences.

Third, we provide indirect measures of disconnection based on probabilistic tractography in healthy participants. Even though we did not directly measure differences in tract integrity in LG$^+$ and LG$^-$ patients, we argue that not only lesions in the pSTS but also the connected networks are associated with the processing of sound at short timescales. We believe that this interpretation is valid because previous studies have confirmed that behavioral deficits are explained to a similar extent by both the local damage and indirectly measured disconnection (*Salvalaggio et al., 2020*). The specificity of temporo-cerebellar connectivity is further supported by control analyses that indicate that (i) there is no connectivity from another control region in the temporal-parietal cortex and (ii) motor cortex-cerebellar connectivity shows different trajectories and cerebellar terminations (Materials and methods and Results – Control analysis, *Figure 3—figure supplement 1*, *Figure 4—figure supplement 1*, *Figure 4—figure supplement 2*, and *Figure 4—figure supplement 3*). Future research in stroke patients is necessary to test for actual changes in temporo-cerebellar fibers (e.g., alterations in fractional anisotropy [FA]) to establish a direct link between impaired processing of sound at short timescales and tract integrity.

## Conclusions

We show that the left posterior temporal cortex contributes to audition in a time-sensitive manner. This functional characteristic, identified by lesion-symptom mapping, has led to the discovery of a specific cortico-subcortical structural connectivity pattern. Taken together, these results provide compelling evidence for a mechanism that in its simplicity not only applies to audition but may extend to other modalities relying on similar structural connectivity patterns (visual motion perception [*Sokolov et al., 2014*]; multisensory integration [*Baumann and Greenlee, 2007*; *Schmahmann and Pandya, 1991*]). This points to a possible common neurobiological function (*Schmahmann, 2004*) supporting internal modeling of a dynamic environment.

# Materials and methods

## Participants

A group of stroke patients (lesion group, LG, *N* = 12) and a group of healthy controls (control group, CG, *N* = 12) closely matched for handedness, gender, age, and formal education (*Supplementary file 1*) were selected from databases at the Leipzig University Hospital Day Clinic for Cognitive Neurology and the Max Planck Institute for Human Cognitive and Brain Sciences, Leipzig, Germany. For patients (*Supplementary file 1*), initial inclusion criteria were a chronic ischemic stroke (time since lesion ≥12 months) in the left temporal lobe, no previous cerebral infarctions, or lesions to other areas of the brain, and no history of other neurological or psychiatric disorders. To confirm normal hearing for their respective ages, patients and controls underwent audiometric screening with air conduction inside a sound-proof cabin (according to the *Guidelines for Manual Pure-Tone Threshold Audiometry*, American Speech-Language-Hearing Association, http://www.asha.org/docs/html/GL2005-00014.html) using a computer-based audiometer (MAICO MA 33, MAICO Diagnostic GmbH; headphones MAICO DD45; audiometric test frequencies 125 Hz to 8 kHz). One patient had to be excluded, as hearing thresholds between 0.5 and 2 kHz did not meet the criteria for age-normal hearing (International Standard ISO 7029, 2000). Hearing loss (dB HL) on both ears did not differ significantly between the remaining patients and the healthy controls (0.5 kHz: patients mean ± SD = 10.6 ± 3.7, controls = 12.3 ± 4.5 dB HL, *U* = 56, p = 0.38; 1.0 kHz: patients = 9.4 ± 3.0, controls = 11.7 ± 5.6 dB HL, *U* = 55.5, p = 0.35; 1.5 kHz: patients = 11.9 ± 4.0, controls = 13.3 ± 8.3 dB HL, *U* = 70.5, p = 0.93; 2 kHz: patients = 12.9 ± 7.7, controls = 11.5 ± 7.0 dB HL, *U* = 78.5, p = 0.71). All participants were German native speakers and right-handed (handedness index score >40). Time since lesion varied from 12 to 150 months (*M* = 60, SD = 42.1 months). Stroke severity was assessed by means of the National Institute of Health Stroke Scale (NIHSS, http://www.nihstrokescale.org/; German translation and validation [*Berger et al., 1999*]). The NIHSS (score range 0–42) provides a simple protocol to assess stroke-related motor (e.g., paralysis), non-motor, and cognitive functions (e.g., consciousness, presence of aphasia, or neglect). Although typically applied to patients suffering from acute stroke, the overall

NIHSS scores for the current (chronic) patient group ($M$ = 1.8 ± 1.0, range 0–4) reflect the severity of residual neurological deficits, ranging from normal function (score = 0) to minor impairments (scores = 1–4). However, NIHSS test items for language functions may not capture some residual deficits in chronic stroke patients. Therefore, language functions were assessed using the Aachener Aphasie Test (AAT) (*Huber et al., 1984*; *Supplementary file 1*).

For structural connectivity analysis, an additional group of 12 healthy participants was selected from the same databases. Individual patient-control pairs were matched in terms of age (51.8 ± 9.25 years), gender (seven male), and handedness (handedness index 83.8 ± 15.5; Edinburgh Handedness Inventory; *Oldfield, 1971*), to control for potential effects of these factors on FA values (*Pal et al., 2011*; *Powell et al., 2012*; *Salat et al., 2005*).

All experimental procedures were approved by the local ethics committee of the University of Leipzig according to the Declaration of Helsinki and written-informed consent was given by each participant. All participants were naïve to the objective of the experiments and were financially compensated for their time and travel costs.

## Stimuli and tasks

Speech stimuli (word, non-word, and phoneme pairs) were spoken in a soundproof cabin by a professionally trained female German native speaker, digitized with 16-bit resolution at a sampling rate of 44.1 kHz stereo using AlgoRec 2.1 (Algorithmix GmbH, Waldshut-Tiengen, Germany) and subsequently converted to mono. Offline editing consisted of cutting at zero crossings before and after each word, normalization to an average intensity of 70 dB using the PRAAT software (http://www.praat.org/; *Boersma and Weenink, 2001*). Non-speech stimuli were synthesized using Audacity (http://www.audacity.sourceforge.net/). The complex tones consisted of two 1000 and 2000 Hz ($\Delta f$ = 1000 Hz) sinusoidal components with an average intensity of 70 dB and rise-plateau-decay values of 7-ms-10-ms-7-ms (*Wright, 1960*). These parameters and the stimulus presentation mode are based on previous work and allow for threshold determination in clinical populations (*Fink et al., 2005*). All speech and non-speech stimuli were presented binaurally at a fixed intensity level via headphones (Sennheiser HD 202).

Individual threshold levels for *auditory temporal order judgments* (task 1) and *discrimination of complex tones (micropatterns)* (task 2) were determined to evaluate participants' perceptual abilities to process non-verbal auditory spectro-temporal information. For task 1, auditory temporal order thresholds were defined as the shortest SOA at which participants can correctly judge the temporal order of the two consecutively presented tone components (component A following B or B following A). For task 2, auditory discrimination thresholds referred to the shortest SOA at which participants can judge the difference between two complex tones (AB sounds perceptually different from BA but identical to AB). Participants were familiarized with the tones during a training period to ensure that they were readily audible to allow for order and perceptual difference judgments to be obtained. Participants completed a forced-choice decision for both tasks with the brief randomly presented complex tones, separated by SOAs ranging from 1000 ms (suprathreshold level for all participants) to 2 ms. For auditory order threshold determination (task 1) participants indicated whether the pitch of the first was higher or lower than the second component of the complex tone. A 50 ms-down 5 ms-up fixed step size staircase procedure with an SOA decrease after three correct and an increase after an incorrect response (reversal) was used. For discrimination thresholds (task 2) the complex tones were randomly presented below the individual temporal order threshold as determined by the previous test. Each presentation consisted of two complex tones in which the components had either the same (e.g., AB-AB) or reversed temporal order (e.g., AB-BA). Participants indicated whether the two complex tones were perceived as same or different. The same staircase procedure was applied, using fixed step sizes of 20 ms-down and 2 ms-up. The tasks were terminated after eight staircase reversals and auditory order (task 1) and discrimination (task 2) thresholds were calculated as the average of the last five SOAs at the staircase reversals points. This led to a probability of a 79.4 % correct threshold level for auditory discrimination and order judgments (*Levitt, 1971*).

To evaluate basic receptive and discriminative language abilities in the two groups, we used a test for auditory discrimination of words and non-words taken from the German LeMo Test battery (German version of *Lexikon modellorientiert*, model-based assessment of aphasia; *De Bleser et al., 1997*; *De Bleser et al., 2004*). Participants had to judge whether spoken mono-morphemic minimal

pairs (in two separate word and pseudoword lists) were identical or not (e.g., words: Dach [dax] (engl. *roof*) – Bach [bax] (engl. *stream*); pseudowords: Pach [pax] – Kach [kax]). Each list consisted of 72 items, 36 of which were identical. Non-identical items varied in terms of their consonant features in either place (e.g., Bauk [bauk] – Baup [pauk]) or manner (e.g., Korf [korf] – Korm [korm]) of articulation. Pseudoword and word lists were presented separately to each participant.

Further language testing involved consonant feature related *discrimination of phoneme pairs* (unpublished material, Day Clinic for Cognitive Neurology, Leipzig). A total of 56 consonant clusters (e.g., /tr/ and /pr/) or single consonants (e.g., /t/ and /p/) that were either identical or varied in terms of their contrastive features were presented. Upon auditory presentation participants were asked to judge whether the presented pairs were the same (21 items) or different (35 items). Phonemic contrasts of non-identical pairs resulted from differences either in place of articulation (e.g., /p/ and/t/), voicing (e.g., /p/ and /b/), place of articulation and voicing (e.g., /p/ and /g/), or were fricative contrasts (e.g., /f/ and /ʃ/).

A short familiarization period consisting of five additional words, pseudowords, or phoneme pairs preceded the testing phase. Feedback was only provided during the training period and items were not repeated during the testing phase.

## Data acquisition
### Structural and diffusion-weighted imaging
High-resolution anatomical T1- and T2-weighted magnetic resonance (MR) scans suitable for lesion reconstruction were available for all patients. Structural and diffusion-weighted datasets used for probabilistic tractography in healthy elderly participants were acquired with standard imaging protocols.

### Imaging procedures
The high resolution ($1 \times 1 \times 1$ mm$^3$) structural datasets were obtained at 3 T on a Siemens TrioTim (Siemens Healthcare, Erlangen, Germany) or Bruker BioSpin (BioSpin GmbH, Rheinstetten, Germany) MR system with a 32-channel phased-array head array coil using an MP-RAGE sequence (*Mugler and Brookeman, 1990*) with inversion times (TI) of 650 ms, repetition times (TR) of 1.3 s, echo times (TE) of 3.93 ms, flip angles of 10°, an imaging matrix of $128 \times 128$ pixel, and a field of view (FOV) of $256 \times 240 \times 176$ mm$^3$. Additional T2-weighted fluid-attenuated inversion recovery (FLAIR) scans were available for all patients. The LIPSIA software (*Lohmann et al., 2001*) was used to convert DICOM and Bruker datasets into three-dimensional images (voxel size $1 \times 1 \times 1$ mm$^3$) in NIfTI format.

High-resolution diffusion-weighted MR datasets were acquired at a 3 T Siemens TrioTim scanner (32-channel phased-array head array coil, Siemens Healthcare, Erlangen, Germany) with a twice-refocused spin echo EPI sequence (*Reese et al., 2003*) using a TE of 100 ms, a TR of 12 s, a $128 \times 128$ image matrix, an FOV of $220 \times 220$ mm$^2$ with a total of 88 axial slices (no gap) and a resolution of $1.7 \times 1.7 \times 1.7$ mm$^3$. Diffusion weighting was isotropically distributed along 60 diffusion-encoding gradient directions with a *b*-value of 1000 s/mm$^2$. Eight images with no diffusion weighting (*b*0) were acquired initially and interleaved after each block of 10 diffusion-weighted images providing an anatomical reference for offline motion correction. Diffusion-weighted datasets were analyzed using LIPSIA and the FMRIB's diffusion toolbox (FDT, Oxford Centre for Functional Magnetic Resonance Imaging of the Brain Diffusion Toolbox, http://fsl.fmrib.ox.ac.uk/fsldownloads/, FSL version 4.1.9; *Behrens et al., 2003*; *Jenkinson et al., 2012*; *Smith et al., 2004*). T1-weighted structural scans were used for skull stripping and the resulting images were co-registered to Talairach space (*Talairach and Tournoux, 1988*). Motion correction was performed based on the seven reference images without diffusion weighting (*b*0) using rigid-body transformations (*Jenkinson et al., 2002*) as implemented in FMRIBs software library (FSL). Motion correction parameters were interpolated for all volumes and combined with a global registration to the T1-weighted anatomy and gradient direction for each volume was corrected using the rotation parameters. These transformations were applied to all volumes, gradient directions were averaged, and the images were interpolated to an isotropic voxel resolution of 1 mm. Finally, a diffusion model was fitted to the preprocessed diffusion-weighted datasets with Bayesian Estimation of Diffusion Parameters Obtained using Sampling Techniques (FDT's BedPostX), an algorithm allowing for multiple fiber orientations (default value, $N = 2$) within each voxel (*Behrens et al., 2007*; *Behrens et al., 2003*).

## Data analysis

### Language and behavioral data

All statistical analyses were performed using SPSS 20.0 (IBM Corp., 2011). Means and SD were calculated for the respective variables and groups. Shapiro-Wilk normality tests revealed that most of the data violated the assumption of normality for any of the psychometric variables. Therefore, non-parametric test methods were applied. Exact p-values ($\alpha$ = 0.05) are reported for small sample sizes. For independent samples, one-sided Mann-Whitney $U$ tests were used to examine whether error rates or threshold levels were significantly higher in patients than in controls. Friedman tests with subsequent post hoc Wilcoxon signed-rank tests (two-sided) were performed to test for within-group differences in error rates between linguistic stimulus categories (words, pseudowords, and phonemes) or contrastive features (place or manner of articulation for words and pseudowords; place, voicing, place and voicing, or fricatives for phonemes). Bonferroni adjustment was applied if necessary. Associations between different linguistic and non-linguistic stimulus categories were analyzed by means of one-sided Spearman's rank-order correlations coefficient to test for an increasing (positive) relationship between error rates and threshold levels in patients. Categorical variables were analyzed with Fisher's exact test.

### Lesion mapping and subtraction

Individual lesions were manually delineated on axial slices (slice thickness 1 mm) of T1-weighted images. To improve lesion characterization, lesion mapping was guided by co-registered T2-FLAIR images. The MRIcron software (http://www.mccauslandcenter.sc.edu/mricro/mricron/) was used to create a binary lesion map (volume of interest [VOI]) for each subject, to generate individual lesion masks for normalization, and to estimate lesion volumes (*Supplementary file 1*). Individual T1-weighted images, co-registered T2-FLAIR, and lesion maps were spatially normalized to standard stereotaxic Montreal Neurological Institute (MNI) space by means of the unified segmentation approach (*Ashburner and Friston, 2005*; *Crinion et al., 2007*) using *Clinical toolbox* (*Rorden et al., 2012*) in SPM8 (Wellcome Department of Imaging Neuroscience, London, http://www.fil.ion.ucl.ac.uk/spm). This toolbox provides an MR imaging (MRI) template and spatial priors for elderly participants. Cost-function masking was applied during normalization to achieve optimal anatomical co-localization of brain structures (*Andersen et al., 2010*). All neuroanatomical data related to individual lesions, as well as lesion overlap, and subtraction plots are reported in MNI space. Anatomical specification was based on visual inspection along with the macro-anatomical labels provided by the SPM-based Anatomy Toolbox (*Eickhoff et al., 2005*). For visualization of overall lesion distribution, individual lesion maps were superimposed (lesion overlap) on the scalp-stripped mean patient T1-weighted image (SPM8, ImCalc). Subsequent subtraction analysis was aimed at linking language and behavioral deficits related to representation of auditory temporal information to the anatomy of brain tissue damage (lesion-symptom mapping). These subtraction analyses account for differences between brain regions specifically contributing to a certain function and more vulnerable, but commonly damaged ones. This is achieved by contrasting lesions of patients with and without a (behavioral) deficit of interest at a certain behavioral cut-off value (*Liebermann et al., 2013*; *Rorden and Karnath, 2004*; *Rorden et al., 2007*). Raw data from each subtest were transformed into *z*-scores corresponding to the controls mean and SD to group patients by deficit (impaired performance) based on behavioral data (*Liebermann et al., 2013*). Patients performing outside two SD of the healthy control groups mean in at least two of the speech (discrimination of word, pseudoword and phoneme pairs) or non-speech (auditory order and MP discrimination thresholds) subtests were assigned to the group with impaired performance (deficit-positive lesion group, LG+). Remaining patients were assigned to the control patient group performing within the normal range (two SD of the control groups mean) on speech and non-speech subtests in the same set of experiments (deficit-negative lesion group, LG−). Groups (LG+ and LG−) were compared for differences in age, gender, handedness, education level, lesion volume, time since lesion and performance on the Token test using independent sample *t*-tests. Subtraction plots were estimated from individual VOIs using MRIcron (http://www.mccauslandcenter.sc.edu/mricro/mricron/). The resulting subtraction plots indicate the relative frequency (percentage) of overlapping lesions in the patient group with abnormal performance (LG+) after subtracting the lesion overlap of LG− from the overlap of LG+. To infer statistical significance of differences in lesion distribution between LG+ and LG−, the non-parametric Liebermeister test was applied to binomial

data that reflected group classification. Non-parametric mapping (distributed with MRIcron) was used for voxelwise statistical analysis including only voxels affected in at least one patient and correcting for multiple comparisons by permutation testing ($N$ = 12, permutations = 4000) (*Rorden et al., 2007*).

## Lesion analysis-based probabilistic diffusion tractography

Probabilistic fiber tracking was performed from regions significantly more frequently affected in LG$^+$ as compared to LG$^-$ to localize regions contributing to the deficit in view of intact white matter fiber tracts in elderly subjects. These VOIs (*Figure 5A*) were affinely transformed from MNI to each participant's diffusion space (FMRIB's Linear Image Registration Tool, FLIRT, 12 degrees of freedom; *Jenkinson et al., 2002*; *Jenkinson and Smith, 2001*). Crossing fiber probabilistic tractography (FDT ProbTrackX) with analysis parameters (step length = 0.5 mm, number of steps = 2000, number of pathways = 5000, curvature threshold = 0.2) previously used to study cerebro-cerebellar-cortical tracts (*Salmi et al., 2010*) was performed based on each participant's probability distributions of voxelwise principal diffusion directions (FDT BedPostX). This algorithm (*Behrens et al., 2007*; *Behrens et al., 2003*) computes the sum of connectivity distributions by generating streamlines from each function-ally informed seed mask passing through the respective other mask but excluding pathways that cross into the right hemisphere (sagittal midline exclusion mask at the level of the corpus callosum). Indi-vidual patterns of structural connectivity were subdivided into separate bundles by manually placing several inclusion (waypoint) masks to subsequently classify anatomically defined white matter tracts connecting the identified brain regions to cortical and subcortical areas. These left hemisphere inclu-sion masks were based on the unrestricted overall connectivity pattern of all subjects (crossing fiber probabilistic tractography from both seeds only including a midline exclusion mask) and the ICBM DTI-81 white matter label atlas (http://www.loni.usc.edu; *Mori et al., 2008*). Masks were placed coro-nally at the level of the left and right MCP (cortico-ponto-cerebellar tract) and SCP (cerebello-rubro-thalamic tract; *Granziera et al., 2009*; *Jissendi et al., 2008*), in the left periventricular white matter lateral to the superior corona radiata (SLF; *Makris et al., 2005*), in the left anterior floor of the external capsule (IFOF; *Catani et al., 2002*) and in the left posterior corona radiata above the roof of the lateral ventricle (dorsal subcomponent of the IFOF or MLF; *Makris et al., 2013a*; *Martino et al., 2010*). All masks were reverse-normalized from MNI standard to individual diffusion space (FLIRT, 12 degrees of freedom). Correct locations of seeds, exclusion and inclusion masks were confirmed visually in native space. All estimated connectivity distributions were scaled across subjects by dividing individual white matter tracts by the total number of probabilistic streamlines to account for differences between tracts due to differences in normalized seed voxel sizes and masks. The tracts were then thresholded to include only voxels that received at least $1 \times 10^{-7}$ % of the scaled total number of streamlines sent out from the seed masks (samples per voxel [5000] multiplied by the number of voxels in the seed masks [mean = 827.2 ± 61.2 (SD)]; *Rilling et al., 2008*). Thresholded individual tractography results were binarized, transformed into standard MNI space, and averaged to display group variability maps, to quantify the overlap in tract topography. These maps indicate the degree of spatial variability and overlap in each voxel. The pathways and terminations identified were compared against anatom-ical pathways as defined in primate and human brain dissections or MRI-based anatomical atlases of cortical and subcortical gray or white matter (Oxford thalamic connectivity atlas [*Behrens et al., 2003*], MNI Talairach atlas [*Lancaster et al., 2007*; *Lancaster et al., 2000*], probabilistic cerebellar atlas [*Diedrichsen et al., 2009*], Harvard-Oxford cortical and subcortical probability maps available from FSL [*Jenkinson et al., 2012*]).

## Control analysis

The subtraction of LG$^-$ patients from LG$^+$ patients revealed that in LG$^+$ lesions in the left pSTS and MTG were 90 % more frequent compared to LG$^-$ (*Figure 3A*). The reverse contrast (subtraction of LG$^+$ patients from LG$^-$ patients) showed that lesions in the left inferior parietal lobe (IPL) and angular gyrus (AG) (MNI –32 –53 38) and in the most posterior parts of the MTG (MNI –43 –65 19) were 50 % more frequent in LG$^-$ compared to LG$^+$ (*Figure 3—figure supplement 1A*). This difference was not statisti-cally significant ($z$ = –1.81, with critical $z$-value of –2.83 corresponding to permutation FWE-corrected $\alpha$-level of $p < 0.05$). Yet, these regions (IPL, AG, pMTG) taken from the same study population served as negative control seed regions (*Figure 3—figure supplement 1B*) to test for deficit specificity of the identified temporo-cerebellar networks. To further test for the specificity of cerebellar terminations,

we chose another control region in the motor cortex approximately corresponding to the left foot motor area. Based on anatomical landmarks, the spherical VOI (10 mm) was placed at the termination of the central sulcus in the uppermost portion of precentral gyrus and close to the midline of the brain (*Figure 3—figure supplement 1C*; *Lehericy et al., 2000*). The foot motor area was chosen because of the somatotopic organization of corticopontine fibers such that foot fibers run in the most posterior portions of the posterior limb of the internal capsule and in the lateral portions of the cerebral peduncle (*Pan et al., 2012*). This anatomical arrangement was considered to also control for the possibility that adjacent pyramidal fiber tracts were traced in error.

Control analyses paralleled the original analysis steps, including seed transformation from MNI to each participant's diffusion space, probabilistic tractography with the same inclusion masks in the SLF, the MCP and SCP, scaling of tracts, and transformation into standard MNI space.

## Acknowledgements

This work was supported by DFG KO 2268/6-1 granted to SAK; AS was supported by a dissertation award provided by the University of Leipzig, Germany.

## Additional information

### Funding

| Funder | Grant reference number | Author |
| --- | --- | --- |
| Deutsche Forschungsgemeinschaft | DFG KO 2268/6-1 | Sonja Kotz |
| Leipzig University | Dissertation award | Anika Stockert |
| Max-Planck-Gesellschaft | | Sonja Kotz |

The funders had no role in study design, data collection and interpretation, or the decision to submit the work for publication.

### Author contributions

Anika Stockert, Conceptualization, Data curation, Formal analysis, Investigation, Methodology, Visualization, Writing – original draft, Writing – review and editing; Michael Schwartze, David Poeppel, Conceptualization, Writing – original draft, Writing – review and editing; Alfred Anwander, Formal analysis, Methodology, Writing – review and editing; Sonja A Kotz, Conceptualization, Funding acquisition, Methodology, Project administration, Resources, Supervision, Writing – original draft, Writing – review and editing

### Author ORCIDs

Anika Stockert http://orcid.org/0000-0001-5804-2498
Michael Schwartze http://orcid.org/0000-0003-3366-4893
Alfred Anwander http://orcid.org/0000-0002-4861-4808
Sonja A Kotz http://orcid.org/0000-0002-5894-4624

### Ethics

Human subjects: The protocol of the current research was approved by the ethics committee of the University of Leipzig, Germany (Protocol Number: 953). All participants provided written, informed consent before the start of data collection.

### Decision letter and Author response

Decision letter https://doi.org/10.7554/eLife.67303.sa1
Author response https://doi.org/10.7554/eLife.67303.sa2

## Additional files

### Supplementary files
- Transparent reporting form
- Supplementary file 1. Supplementary patient data, methods and control analysis.

### Data availability
There is restricted access to the data due to German legal regulations of patient protection. We have made all data which we can legally share accessible via figshare (https://doi.org/10.6084/m9.figshare. 14213393). Here we have provided all data (lesion data, scripts, behavioral data that allowed lesion-symptom mapping) for reproduction of the critical seed region for a tracking analysis. Anonymisation of MRI/DTI data is not allowed either through the ethics agreement nor the participants' consent. We have made a clear statement that we seek open dialogue about how we have analysed our data. Further, given the data that we have provided, any interested researcher can (1) approach us about our analysis, (2) can take a set of open source age-matched structural MRI/DTI data to replicate our results.

The following dataset was generated:

| Author(s) | Year | Dataset title | Dataset URL | Database and Identifier |
|---|---|---|---|---|
| Stockert A, Schwartze M, Poeppel D, Anwander A, Kotz SA | 2021 | Temporo-cerebellar connectivity | https://doi.org/10.6084/m9.figshare.14213393 | figshare, 10.6084/m9.figshare.14213393 |

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
