## [Decision Letter]

**Acceptance summary:**

This work nicely explores the role of temporo-cerebellar connectivity on auditory processing. The use of lesion-guided connectivity is innovative, and the connectivity demonstrated between stroke areas associated with timing deficits and the cerebellum is interesting in and of itself. The control analyses are helpful and strengthen the conclusions.

**Decision letter after peer review:**

Thank you for submitting your article "Temporo-cerebellar connectivity underlies timing constraints in audition" for consideration by *eLife*. Your article has been reviewed by 3 peer reviewers, including Timothy D Griffiths as Reviewing Editor and Reviewer #1, and the evaluation has been overseen by Barbara Shinn-Cunningham as the Senior Editor.

Essential revisions:

1) The referees all felt that the language in the manuscript needed to be more moderate in terms of the extent to which the link between AST and a further role for the cerebellum is supported.

2) Further control analyses are suggested in the detailed referee comments below which we would like the authors to at least comment on.

*Reviewer #1 (Recommendations for the authors):*

I raise some experimental issues below and also issues related to the interpretation and exposition which I think goes some way beyond the data.

1. The manuscript is written as if asymmetric sampling in time (AST) is a given. The phrase 'widely tested' is correct, but not illuminating.

2. I do not agree that the work 'supports the hypothesis that encoding and modeling of spectro-temporal information relies on a temporo-cerebellar system'. The data show that lesion-related deficits in spectrotemporal analysis occur in regions of the temporal lobe that are connected to the cerebellum.

3. Was a control analysis carried out to assess connectivity from areas of the temporal lobe outside of the areas associated with the temporal deficit?

4. A perfect experiment would have examined right temporal lobe deficits (which the authors would hypothesise to be in the analysis of longer time windows) and assessed the connection of these to the cerebellum.

5. There is a lot of discussion of forward models in the manuscript but the data do not test this. They assess connectivity between areas associated with a temporal deficit in the temporal lobe and the cerebellum, which has been implicated as a basis for forward modelling. I also think that the second conclusion that the work 'suggests a generalizable role of cerebellar-mediated internal models that extends beyond motor control to auditory perception' is not directly justified by the data.

6. Figure 6 suggests an interesting hypothesis that extends the asymmetrical system for processing different time windows in the two hemispheres based on AST in temporal cortex to the contralateral cerebellar hemispheres. Do the data support this hypothesis? As far as I can see the projections from the seed region to cerebellum are bilateral and I do not think any explicit test is carried out to examine the lateralisation of projections to the cerebellum.

In summary, I am not sure that the claimed system for fine temporal analysis involving left posterior temporal lobe and right cerebellum has been clearly demonstrated and would be interested in the authors comments. I do not think the authors need to address point 4 as a condition of publication. Overall, in any event, I felt that the manuscript could be more circumspect.

*Reviewer #2 (Recommendations for the authors):*

The paper is very well written and interesting and a true pleasure to read, when I wasn't getting frustrated that strong conclusions were being drawn based on weak evidence. I wonder whether, given the somewhat tenuous nature of the novel evidence that you provide (with regard to your conclusions), this might be better received as a theory piece with a bit of new empirical data to support the theory, and to motivate new research. I also noted that there was no consideration of data in the literature that failed to support the proposed model, but perhaps this is because there is none.

Control conditions would have helped to bolster your case, including a group of control lesion patients (e.g., right hemisphere homologous lesions to see if the hemispheric differences in temporal processing hold up; another secondary sensory region to show a dissociation between type of lesion and type of behavioral deficit) and control tasks to show that deficits are specific to hypothesized function of the lesioned area (e.g., that they do not simply represent that lesion patients have cognitive impairment). Establishing a functional connection between the temporal lobe and cerebellar activity during temporally informed/based auditory tasks would also help to strengthen the evidence for your claims.

*Reviewer #3 (Recommendations for the authors):*

I only have a few queries in addition to the public review comments.

1) Were further analyses attempted to investigate the relationship between behavioural auditory temporal analysis and the structural integrity of the identified pathways. I recognise that the numbers are small for this type of analysis, but it could be exploratory.

2) Was tracking attempted for the stroke participants who did not have lesions to the region of interest? Did these participants demonstrate intact behaivour and intact cerebellar pathways?

3) I found the statistical results of the phoneme discrimination data on page 9 somewhat difficult to follow.

---

## [Author Response]

Essential revisions:1) The referees all felt that the language in the manuscript needed to be more moderate in terms of the extent to which the link between AST and a further role for the cerebellum is supported.

Following your advice and that of the referees, we now refrain from the use of asymmetric sampling per se and only refer to the fact that the data indirectly address ideas raised in the context of this hypothesis.

2) Further control analyses are suggested in the detailed referee comments below which we would like the authors to at least comment on.

When conceptualizing the study we certainly considered a right-hemisphere patient control group. However, after careful consideration we decided against this option for the following reason: based on a thorough screening of an extensive patient database, we were unable to identify enough patients with comparable right-hemisphere stroke lesions (both in terms of location and size). From a clinical point of view this is likely because right temporal stroke lesions tend to be much more extensive in size before patients become symptomatic.

Alternatively, we now chose to run two control analyses. First, based on the reverse contrast of the subtraction analysis, we defined the lesion site of the LG- relative to the LG+ patient group (Figure 3—figure supplement 1). This resulted in regions in the inferior parietal lobule (IPL)/angular gyrus (AG)/most posterior MTG (pMTG) in the LG- group (Figure 3—figure supplement 1) . These regions served as negative control seed regions to test for the deficit specificity of the identified temporo-cerebellar networks. Second, to test for the specificity of cerebellar terminations, we chose another control region in the motor cortex approximately corresponding to the left foot motor area (Figure 3—figure supplement 1). We choose the foot area, because of the somatotopic organization of corticopontine fibers such that foot fibers run in the most posterior portions of the posterior limb of the internal capsule and in the lateral portions of the cerebral peduncle (Pan et al., 2012). This anatomical arrangement was considered to control for the possibility that adjacent pyramidal fiber tracts were traced in error.

The results of these analyses are as follows:

“Fibers originating from the seed masks in the control region (left IPL/AG, pMTG) were identified in periventricular white matter. […] These results further demonstrate the specificity of temporo-cerebellar and thalamo-temporal projections in relation to impaired processing of sound at short timescales.”

Reviewer #1 (Recommendations for the authors):I raise some experimental issues below and also issues related to the interpretation and exposition which I think goes some way beyond the data.1. The manuscript is written as if assymetric sampling in time (AST) is a given. The phrase 'widely tested' is correct, but not illuminating.

We have removed reference to asymmetric sampling in time (AST) throughout the manuscript.

2. I do not agree that the work 'supports the hypothesis that encoding and modeling of spectro-temporal information relies on a temporo-cerebellar system'. The data show that lesion-related deficits in spectrotemporal analysis occur in regions of the temporal lobe that are connected to the cerebellum.

We appreciate this specification and have rephrased this statement according to the referee’s suggestion. See page 15 of the revised manuscript:

“The evidence we describe (i) shows that lesion-related deficits in spectrotemporal analysis occur in posterior temporal regions connected to the cerebellum.”

Further, we motivate and explain the rationale of a functional interpretation of the applied method of an indirect estimation of disconnection based on healthy participants and lesion data. See pages 6 and 12 of the revised manuscript:

“Assessing connectivity in healthy participants based on lesion information is a relatively new method that measures structural disconnection in networks associated with given anatomical regions (Foulon et al., 2018). This allows for the indirect estimation of the lesion effect on structural brain networks. In this regard, it was shown that behavioral deficits can be explained similarly by local brain damage and indirectly measured disconnection (Salvalaggio et al., 2020).”

“We next used the respective areas as seed regions for probabilistic fiber tractography in a healthy age-matched sample to visualize the underlying common connectivity pattern (see Methods). Thus, we indirectly explored the association between posterior superior temporal disconnection and processing of sound at short timescales.”

We are aware that this kind of interpretation of an indirect measure could be considered as a potential limitation. We therefore added a statement in a new limitations section. See page 22 of the revised manuscript:

“Third, we provide indirect measures of disconnection based on probabilistic tractography in healthy participants. […] Future research in stroke patients is necessary to test for actual changes in temporo-cerebellar fibers (e.g., alterations in fractional anisotropy) to establish a direct link between impaired processing of sound at short timescales and tract integrity.”

3. Was a control analysis carried out to assess connectivity from areas of the temporal lobe outside of the areas associated with the temporal deficit?

In the revised version of the manuscript, we now include two control analyses. As suggested by the referee, we choose an area in the temporal(-parietal) lobe not associated with temporal deficits as well as an independent area (M1) that, based on prior knowledge, is expected to show cerebellar connectivity. First, based on the reverse contrast of the subtraction analysis, we defined the lesion site of the LG- relative to the LG+ patient group (Figure 3—figure supplement 1). This resulted in regions in the inferior parietal lobule (IPL)/angular gyrus (AG)/most posterior MTG (pMTG) in the LG- group (Figure 3—figure supplement 1). These regions served as negative control seed regions to test for the deficit specificity of the identified temporo-cerebellar networks. Second, to test for the specificity of cerebellar terminations, we chose another control region in the motor cortex approximately corresponding to the left foot motor area (Figure 3—figure supplement 1). We choose the foot area, because of the somatotopic organization of corticopontine fibers such that foot fibers run in the most posterior portions of the posterior limb of the internal capsule and in the lateral portions of the cerebral peduncle (Pan et al., 2012). This anatomical arrangement was considered to control for the possibility that adjacent pyramidal fiber tracts were traced in error.

Further details on the control analyses and respective results are reported in the Supplementary Information.

The results of these analyses are as follows:

“Fibers originating from the seed masks in the control region (left IPL/AG, pMTG) were identified in periventricular white matter. […] These results further demonstrate the specificity of temporo-cerebellar and thalamo-temporal projections in relation to impaired processing of sound at short timescales.”

4. A perfect experiment would have examined right temporal lobe deficits (which the authors would hypothesise to be in the analysis of longer time windows) and assessed the connection of these to the cerebellum.

When conceptualizing the study, we certainly considered a right-hemisphere patient control group. However, after careful consideration we decided against this option for the following reason: based on a thorough screening of an extensive patient database, we were unable to identify enough patients with comparable right-hemisphere stroke lesions (both in terms of location and size). From a clinical point of view this is likely because right temporal stroke lesions tend to be much more extensive in size before patients become symptomatic.

As this can be considered a limitation to our study, we discuss this point in the new limitations section. See page 21 of the revised manuscript:

“Second, although the results confirm the processing of sound at short timescales in the left STS in patients with left temporal lesions, we did not include a right hemisphere patient control group to test for lateralization. This would be problematic in the first place as right hemisphere lesions tend to be more extensive and rarely spare the primary auditory cortex. While a comparison of left and right temporal lesions would have allowed distinguishing processing differences of shorter and longer timescales such a comparison would have been likely confounded by a primary auditory processing deficit. Future studies could overcome this problem by using a virtual lesion approach (i.e., by applying inhibitory transcranial magnetic stimulation) that would allow for reversible deactivation of left and right posterior STS to test for verbal and non-verbal processing differences.”

5. There is a lot of discussion of forward models in the manuscript but the data do not test this. They assess connectivity between areas associated with a temporal deficit in the temporal lobe and the cerebellum, which has been implicated as a basis for forward modelling. I also think that the second conclusion that the work 'suggests a generalizable role of cerebellar-mediated internal models that extends beyond motor control to auditory perception' is not directly justified by the data.

We agree that given the current evidence, this conclusion may not be entirely justified. However, we think an integrative interpretation of the results with respect to a generalizable role of the cerebellum in forward modelling is important. Therefore, we now clarify in the introduction that cerebellar mediated internal models serve as a ‘theoretical interpretation’. See page 6 of the revised manuscript:

“An integrative theoretical interpretation of the predicted results from the perspective of a cerebellar temporal cortex interface – with potential lateralization reflecting differential temporal sensitivities - offers an intriguing new perspective to explore the anatomical basis of cerebellar internal modeling in motor control and audition, providing a computational generalization that may offer useful new angles for experimentation.”

and rephrased the conclusions accordingly. See page 15 of the revised manuscript:

“The evidence we describe […] is in line with the concept of a generalizable role of cerebellar-mediated internal models that extends beyond motor control to auditory perception.”

We hope these modifications make it clear for the reader that we offer a perspective on the role of cerebellar-mediated internal models beyond motor control that invites further investigation.

6. Figure 6 suggests an interesting hypothesis that extends the asymmetrical system for processing different time windows in the two hemispheres based on AST in temporal cortex to the contralateral cerebellar hemispheres. Do the data support this hypothesis? As far as I can see the projections from the seed region to cerebellum are bilateral and I do not think any explicit test is carried out to examine the lateralisation of projections to the cerebellum.

We agree with the referee that we did not explicitly test lateralization in the current study. In fact, as pointed out by the referee, we not only report cross-lateral but also ipsilateral cortico-cerebellar connectivity. We now added a theoretical interpretation of bilateral temporo-cerebellar connectivity in the discussion and elaborated on how this may support lateralization in the STS (page 19). We further changed Figure 6 and its legend accordingly. See page 42 of the revised manuscript.

“Contrary to our original hypothesis, we found both ipsi- and cross-lateral cortico-cerebellar connectivity. In this regard, Boemio and colleagues (2005) showed in an fMRI study that the STS but not the superior temporal gyrus (STG) shows duration-sensitive lateralization for shorter and longer timescales. Based on these findings the authors proposed that the bilateral STS receive input differently from the STG through intra- and interhemispheric fibers, weighting information towards short-timescales of sound processing. Such weighting might be guided by event boundaries across different timescales encoded in both cerebellar hemispheres.”

In summary, I am not sure that the claimed system for fine temporal analysis involving left posterior temporal lobe and right cerebellum has been clearly demonstrated and would be interested in the authors comments. I do not think the authors need to address point 4 as a condition of publication. Overall, in any event, I felt that the manuscript could be more circumspect.Reviewer #2 (Recommendations for the authors):The paper is very well written and interesting and a true pleasure to read, when I wasn't getting frustrated that strong conclusions were being drawn based on weak evidence. I wonder whether, given the somewhat tenuous nature of the novel evidence that you provide (with regard to your conclusions), this might be better received as a theory piece with a bit of new empirical data to support the theory, and to motivate new research. I also noted that there was no consideration of data in the literature that failed to support the proposed model, but perhaps this is because there is none.

We agree that given the current evidence, a strong conclusion with respect to cerebellar-mediated internal models may not be entirely justified. However, we think an integrative interpretation of the results with respect to a generalizable role of the cerebellum in forward modelling is important. In line with the impression of the referee that parts of the manuscript should more describe a theory piece, we edited the manuscript in such a way that the interpretation of data and theoretical implications became clear and distinct.

An integrative theoretical interpretation of the predicted results from the perspective of a cerebellar temporal cortex interface – with potential lateralization reflecting differential temporal sensitivities - offers an intriguing new perspective to explore the anatomical basis of cerebellar internal modeling in motor control and audition, providing a computational generalization that may offer useful new angles for experimentation.

“The evidence we describe (i) shows that lesion-related deficits in spectrotemporal analysis occur in posterior temporal regions connected to the cerebellum […] and is in line with the concept of a generalizable role of cerebellar-mediated internal models that extends beyond motor control to auditory perception.”

Regarding the referee’s last comment, there is – to the best of our knowledge – no data that failed to support our model.

Control conditions would have helped to bolster your case, including a group of control lesion patients (e.g., right hemisphere homologous lesions to see if the hemispheric differences in temporal processing hold up; another secondary sensory region to show a dissociation between type of lesion and type of behavioral deficit) and control tasks to show that deficits are specific to hypothesized function of the lesioned area (e.g., that they do not simply represent that lesion patients have cognitive impairment). Establishing a functional connection between the temporal lobe and cerebellar activity during temporally informed/based auditory tasks would also help to strengthen the evidence for your claims.

We appreciate the referee’s concerns regarding control groups and conditions, also shared by the HE and referee 1. Please accept the same answers that we provided previously in response to your concerns.

When conceptualizing the study, we certainly considered a right-hemisphere patient control group. However, after careful consideration we decided against this option for the following reason: based on a thorough screening of an extensive patient database, we were unable to identify enough patients with comparable right-hemisphere stroke lesions (both in terms of location and size). From a clinical point of view this is likely because right temporal stroke lesions tend to be much more extensive in size before patients become symptomatic.

Alternatively, we now chose to run two control analyses. First, based on the reverse contrast of the subtraction analysis, we defined the lesion site of the LG- relative to the LG+ patient group (Figure 3—figure supplement 1). This resulted in regions in the inferior parietal lobule (IPL)/angular gyrus (AG)/most posterior MTG (pMTG) in the LG- group (Figure 3—figure supplement 1). These regions served as negative control seed regions to test for the deficit specificity of the identified temporo-cerebellar networks. Second, to test for the specificity of cerebellar terminations, we chose another control region in the motor cortex approximately corresponding to the left foot motor area (Figure 3—figure supplement 1). We choose the foot area, because of the somatotopic organization of corticopontine fibers such that foot fibers run in the most posterior portions of the posterior limb of the internal capsule and in the lateral portions of the cerebral peduncle (Pan et al., 2012). This anatomical arrangement was considered to control for the possibility that adjacent pyramidal fiber tracts were traced in error.

Regarding a possible control task showing that the differences between LG+ and LG- groups we observed are not solely based on cognitive impairment, we did not include such tasks in the study protocol. However, the clinical files of all patients included results of a comprehensive neuropsychological test battery. Based on these neuropsychological test results we can conclude that there are no differences in cognitive impairment between the two groups (deficit positive patients, LG+, are marked with an asterisk in the table). In particular regarding alertness that could have impaired task performance, we found no difference between the two patient groups.

This information was further added to the Supplemental Information.

“Impairments in other cognitive domains (attention, memory, executive function) were present in some patients, but not exclusively in those who belonged to LG+ (Table S7).”

Reviewer #3 (Recommendations for the authors):I only have a few queries in addition to the public review comments.1) Were further analyses attempted to investigate the relationship between behavioural auditory temporal analysis and the structural integrity of the identified pathways. I recognise that the numbers are small for this type of analysis, but it could be exploratory.

Unfortunately, DTI patient data was not available. Therefore, the only possible analysis was using indirect measures in healthy participants to estimate the effect of the lesions associated with impaired processing on the short-timescale of sound on brain networks. The rationale for a functional interpretation of indirect measures of disconnection is given above. No further analysis of subgroups, for example patients that have deficits in processing non-verbal vs. verbal information, was attempted, because essentially the same patients contributed to these subgroups. This is also reflected by a correlation between error rates for verbal measures and threshold levels. See revised manuscript page 10.

2) Was tracking attempted for the stroke participants who did not have lesions to the region of interest? Did these participants demonstrate intact behaivour and intact cerebellar pathways?

Initially we did not include this tracking in the submitted manuscript. We now implemented this analysis as a control analysis to demonstrate the specificity of the results, namely that temporo-cerebellar disconnection is specific to patients with impaired processing of sound at short timescales. In contrast, regions more frequently affected in patients without this deficit (LG-) show no temporo-cerebellar connectivity (i.e., no disconnection).

Further details on the control analyses and respective results are reported in the Supplementary Information. Pages 9-12, Figure 3—figure supplement 1 and Figure 4—figure supplement 1-3.

3) I found the statistical results of the phoneme discrimination data on page 9 somewhat difficult to follow.

We changed this section to make it more accessible. See revised manuscript page 10.

“Post-hoc Wilcoxon tests (Bonferroni-adjusted significance level at p <.0083) confirmed higher error rates for the discrimination of place of articulation contrasts than for voicing (Z = – 2.521, p = .004, r = .728) in phonemes. The same was true when comparing place of articulation to combined place of articulation and voicing contrast (Z = – 2.536, p = .004, r = .732), as well as for place of articulation compared to fricative contrasts (Z = – 2.555, p = .004, r = .738). Similarly, patients showed higher error rates for place of articulation relative to manner of articulation contrasts (Z = -2.684, p = .004, r = .775) in pseudowords.”